# Serum Metabolites Associated with Muscle Hypertrophy after 8 Weeks of High- and Low-Load Resistance Training

**DOI:** 10.3390/metabo13030335

**Published:** 2023-02-24

**Authors:** Denis F. Valério, Alex Castro, Arthur Gáspari, Renato Barroso

**Affiliations:** 1School of Physical Education, University of Campinas, Campinas 13083-970, Brazil; 2Department of Chemistry, Federal University of São Carlos, São Carlos 13565-905, Brazil; 3Brazilian Sport Climbing Association, São Paulo 04616-004, Brazil; 4Sidia Amazon Lab, Sidia Institute of Science and Technology, Manaus 69075-155, Brazil

**Keywords:** muscle hypertrophy, metabolomics, resistance training

## Abstract

The mechanisms responsible for the similar muscle growth attained with high- and low-load resistance training (RT) have not yet been fully elucidated. One mechanism is related to the mechanical stimulus and the level of motor unit recruitment; another mechanism is related to the metabolic response. We investigated the electromyographic signal amplitude (sEMG) and the general metabolic response to high-load RT (HL) and low-load resistance training (LL). We measured muscle thickness by ultrasound, sEMG amplitude by electromyography, and analysis of metabolites expressed through metabolomics. No differences were observed between the HL and LL groups for metabolic response and muscle thickness. A greater amplitude of sEMG was observed in the HL group. In addition, a correlation was observed between changes in muscle thickness of the vastus lateralis muscle in the HL group and levels of the metabolites carnitine, creatine, 3-hydroxyisovalerate, phenylalanine, asparagine, creatine phosphate, and methionine. In the LL group, a correlation was observed between changes in muscle thickness of the vastus lateralis muscle and levels of the metabolites acetoacetate, creatine phosphate, and oxypurinol. These correlations seem to be related to the characteristics of activated muscle fibers, the metabolic demand of the training protocols used, and the process of protein synthesis.

## 1. Introduction

Resistance training (RT) is recognized as an effective means to promote muscle growth [1]. This morphological adaptation is important to improve athletic performance [2] and provides several health benefits, such as the prevention of metabolic diseases [3,4].

Current guidelines recommend that RT should be performed with loads > 60% of a repetition maximum (1RM) to optimize muscle hypertrophy [5,6]. Accordingly, high loads are preferred by researchers [7,8] and practitioners [9]. Conversely, low-load RT, which uses loads around 30% of 1RM, have also been shown to promote muscle growth similar to that seen with high-load RT [10,11,12].

The mechanisms responsible for the similar muscle growth attained with high- and low-load RT have not yet been fully understood. One suggested mechanism is the mechanical stimulus, which is related to the degree of motor unit recruitment [10,13,14]. Traditionally, high-load RT has been assumed to cause greater muscle activation than low-load RT. However, some authors have challenged this assumption and shown that muscle activation is similar in high- and low-load RT [8,13,14]. When performed up (or close) to the concentric failure, low-load RT may recruit motor units similarly to high-load RT [11,15]. The high level of recruitment means that many motor units are being activated and muscle fibers are under intense mechanical stimulation. 

Another proposed mechanism relates to the metabolic response. It has been suggested that the accumulation of metabolites during low-load RT compensates for the lower mechanical tension and stimulates intracellular pathways that induce muscle growth, leading to a hypertrophic response similar to that of high-load RT [13,16].

Although several studies have highlighted metabolite accumulation as a potential mechanism to induce muscle hypertrophy [17,18,19], few have investigated which metabolite is a viable candidate for signaling skeletal muscle hypertrophy in humans [20,21]. In addition, we are unaware of any study that has analyzed the metabolome of low-load RT and compared it with that of high-load RT. Although important, the studies that have analyzed the metabolic response to low-load RT were nevertheless performed with the addition of blood flow restriction, and most of them only investigated the acute metabolic response after a single training session [22,23,24]. Investigating acute responses does not allow inferences on the relationship between the metabolic response to a RT session and hypertrophy observed after a training period. 

Metabolomics represents a promising analysis of the global metabolic response elicited by RT [20,24,25], combining several analytical techniques to identify large amounts of metabolites present in a biological sample [26] and allowing a more comprehensive view of the substrates involved during exercise compared with traditional biochemical techniques focused on isolated compounds.

Thus, this study analyzed muscle activation and the overall metabolic response, using metabolomics, in acute sessions of high- and low-load RT before and after the 8-week training period. This study will contribute to the understanding of the relationship between acute metabolic responses and muscle activation produced by high- and low-load RT sessions and changes in muscle hypertrophy.

## 2. Material and Methods

### 2.1. Participants

Thirty healthy young men, who had not participated in RT programs for at least 12 months before the study, were recruited to the experiments. Individuals with metabolic diseases, such as diabetes, who were on a hypocaloric diet, or who had osteomioarticular problems that could interfere with exercise performance were excluded from the study. Participants with a training frequency of less than 90% attendance in training sessions, or who were absent for more than two consecutive sessions, were also excluded from the study. Participants received information about the benefits, risks, and experimental procedures involved in the research and signed an informed consent form. The procedures were approved by the local Ethics Committee.

Initially, the 30 participants were selected after screening and interview. Five decided to withdraw for personal reasons, and 25 participants were randomly allocated and stratified into two groups, based on vastus lateralis muscle thickness: high- (HL) and low-load (LL). At the end of the training program, 18 participants (age 23 ± 3 years, body mass 67 ± 2 kg, height 172 ± 6 cm) met the minimum training frequency and were considered for the analysis between the HL (*n* = 9) and LL (*n* = 9) groups (Figure 1).

### 2.2. Experimental Design

On the first visit to the laboratory, participants underwent an anthropometric assessment, and ultrasound images (Nanomaxx, Sonosite, Bothell, WA, USA) of the rectus femoris, vastus intermedius, and vastus lateralis muscles were taken.

Participants performed two familiarization sessions with the 1-RM test in the 45° leg press and bilateral leg-extension exercises. After the familiarization sessions, the 1-RM test and re-test were performed for the exercises. There was a minimum of 72 h between laboratory visits. After the 1RM retest, the eight-week training period was started. Participants repeated the 1RM test after the fourth week and 72 h before the last training session to adjust training loads. This procedure was performed to ensure that the participants completed their respective training sessions at the correct intensity, especially during the last training session when the metabolic response was analyzed and muscle activation was assessed.

Blood samples were collected for subsequent metabolic analysis at three time points during the first and last training sessions. Muscle activation of the vastus lateralis was assessed during the three sets in the 45° leg-press exercise.

Finally, ultrasound images were taken 72 h after the eight-week training period (Figure 2). All experimental protocols took place between 8 and 11 am to avoid possible changes in responses to exercise due to the circadian rhythm.

### 2.3. Nutritional Habits

Participants were instructed to maintain the same eating habits in the days before blood sampling. They were familiarized and instructed by experienced researchers to complete the food record of the day before blood collection. Additionally, the total energy intake, amount, and proportions of macronutrients (carbohydrates, lipids, and proteins) filled in the food record were analyzed quantitatively using the NutWin (v. 1.5, Federal University of São Paulo). After a 10 h overnight fast, blood samples were collected from the antecubital vein in serology tubes (Vacuette, 8 mL). Participants also consumed a standardized meal (290 kcal, 60% carbohydrate, 25% lipid, and 15% protein) one hour before baseline blood sampling [24,25].

### 2.4. Maximum Dynamic Strength

Maximum dynamic strength was determined using the one-repetition maximum test (1RM) according to the American Society of Exercise Physiologists (ASEP) guidelines [27] in the leg-press and leg-extension exercises. In this test, the objective was to achieve the maximum weight that could be lifted in one complete movement.

Before the test, participants warmed up for 5 min on an ergometric bike. After the general warm-up, participants performed a specific warm-up consisting of two sets of eight and three repetitions with an estimated load of 50% and 70% of the 1RM, respectively. A three-minute interval was observed between the end of the specific warm-up and the beginning of the test. The tests were performed in the same order for all participants. The initial weight for the test was estimated during the familiarization sessions and was increased until the participant was unable to complete a repetition [27]. The total number of attempts to find the 1RM value was no more than five. There was a three-minute rest interval between attempts, and a 10-minute interval between exercises.

### 2.5. Training Protocol

Participants completed two weekly training sessions using the 45° leg press and bilateral leg-extension exercises, in this order, for eight weeks. The training sessions consisted of a general (5 min on the bike) and a specific warm-up (one set of 10 repetitions at 50% of 1RM), then three sets at an intensity of 80% 1RM for the HL group and 30% of 1RM for the LL group. Both groups performed repetitions to concentric failure with a 90 s interval between sets and a 120 s pause between exercises. Concentric failure was considered when the participant could no longer perform another repetition with the specified range of motion.

### 2.6. Muscle Activation

Muscle activation was assessed by surface electromyography (EMG) obtained only in the 45° leg-press exercise using a 16-channel electromyograph (MP150, Biopac System Inc., Santa Barbara, CA, USA), with the acquisition frequency of the EMG signals set at 1000 Hz and a bandpass filter of 20–500 Hz. Active electrodes (TSD150, Biopac System, Inc., Santa Barbara, USA) with a common mode rejection ratio of >95 dB were used. Before placing the electrodes, the skin was shaved and disinfected with alcohol in the sites designated for application of the electrodes on the vastus lateralis muscle, to reduce the impedance of the skin. The electrodes were placed at the site with the greatest muscle volume identified during an isometric contraction. Positioning was marked with a semi-permanent pen to maintain the positioning. A customized routine (MatLab, The MathWorks, Natick, Middlesex County, MA, USA) was used to digitally filter the raw electromyographic signals (4th order Butterworth, 20–500 Hz bandpass) and convert them to root mean square (RMS).

The RMS of each repetition was normalized by the average RMS of the ten concentric phases of the repetitions completed during the warm-up. The amplitude of the EMG was analyzed in a range of motion of 40°, between 120° and 160°, to reduce the influence of changes in muscle length on the signal. To identify eccentric and concentric phases during the exercise, an electrogoniometer (SG150, Biometrics Ltd., Newport, UK) was attached to the side of the right knee, and the electrogoniometer signal was synchronized with that of the EMG.

### 2.7. Muscle Thickness

A B-mode ultrasound (Nanomaxx, Sonosite, Bothell, WA, USA), with a linear vector probe and a frequency of 7.5 MHz, was used to capture images of the thickness of the rectus femoris, vastus intermedius, and vastus lateralis muscles. Images were taken on the right leg, with images acquired at 50% of the distance between the anterior superior iliac crest and the superior border of the patella [28].

For all measurements, participants were instructed to relax their muscles as much as possible. To ensure the same positioning, participants had the positioning of their bodies delimited on the evaluation stretcher in a standardized manner. The transducer was aligned in the axial plane, perpendicular to the muscles being examined. Images were recorded for later analysis of muscle thickness, using Image J (NIH, Bethesda, MD, USA).

### 2.8. Blood Sampling

Blood collection was performed after participants arrived at the laboratory after fasting for 10 h, consumed the standardized meal, and rested for one hour (collection 0 min). It was repeated at 5 min (+5 min) and at 60 min (+60 min) after the training session [24,25]. Venipuncture was performed via the median cubital vein in the antecubital fossa. After collection, the blood was stored at room temperature for 30 min and then centrifuged at 3000 rpm for 10 min. The serum was stored in a freezer at −80 °C.

### 2.9. Sample Preparation for Metabolomics Analysis

Before the analysis, the 3 kDa filter was washed (Amicon Ultra, Sigma-Aldrich, St. Louis, MO, USA). This wash consisted of 500 µL of Milli-Q H_2_O. After applying Milli-Q H_2_O to the filter, it was centrifuged at 14,000 rpm for 10 min at 4 °C. This process was repeated five times. After the fifth wash, a further spin was performed (filter inversion and rotation of 8000 rpm for 5 s) to eliminate any trace of Milli-Q H_2_O. After the spin, 350 µL of the stored serum was added to the filter, which was centrifuged at 14,000 rpm for 45 min at a temperature of 4 °C. After this period, the solution that had passed through the filter (200 µL) was recovered and placed in a 5 mm MNR tube (Wilmad, Sigma-Aldrich, USA).

To this solution was added 60 µL of 0.1 mol/L phosphate buffer solution (Sodium Phosphate Monobasic, NaH2PO4 0.028 mol/L and Sodium Phosphate Dibasic, Na2HPO4 0.072 mol/L, pH 7.4), 6.06 µL of TSP-d4 (3-(trimethylsilyl)-2,2′,3,3′tetradeuteropropionic acid—Sigma-Aldrich), 50 mmol/L in D2O (D2O, 99.9%; Cambridge Isotope Laboratories Inc., MA, USA) (used as an internal reference), and 340 µl of Milli-Q H_2_O in a 5-mm NMR tube (Wilmad-LabGlass, Vineland, NJ, USA).

### 2.10. Data Acquisition by NMR and Quantification

For the acquisition of spectra, nuclear magnetic resonance spectroscopy (1H-NMR) at 600 MHz (Agilent Technologies Inc., Santa Clara, CA, USA) was used. Data acquisition was performed using VnmrJ software (Varian NMR Systems). The analysis was performed at a constant temperature of 298 K (25 °C). A total of 256 scans were performed, with a delay interval of 1.5 s and an acquisition time of 4 s between each scan. The sample tuning adjustments with the device were performed by a specialized technician, and the other adjustments (lock and shimming) were performed manually for each sample.

Spectral processing, characterization, and quantification of metabolites were performed using Suite 7.6 Chenomx NMR software (Chenomx Inc., Edmonton, AB, Canada). Spectral processing consisted of spectral phase adjustment, baseline correction, and signal removal from water (4.6–5.1 ppm). Then, a 0.3 Hz line-broadening apodization function was applied and the spectrum was calibrated by the internal reference signal (TMSP-d4).

### 2.11. Statistical Analysis

#### 2.11.1. Muscle Thickness, Muscle Activation, and Total Energy Intake of Macronutrients

Data were presented as mean and standard deviation. The normality of data distribution was checked by the Shapiro–Wilk test. For comparisons between and within groups, Mixed Linear Models were used for repeated measures, assuming participants as a random factor, and group (HL and LL) and period (pre- and post-training) as fixed factors for the variables muscle thickness, EMG amplitude, and consumption macronutrients (carbohydrates, lipids, and proteins). Where significant F values occurred, Tukey’s post hoc test was used. The significance level adopted was 5%.

#### 2.11.2. Metabolic Response

To identify the patterns of global metabolic response by the HL and LL groups, a Principal Component Analysis (PCA) was conducted. Before this analysis, to standardize the scale of concentrations between the different metabolites, the autoscaling process was applied [29]. Additionally, a heat map depiction of the metabolite concentrations was generated for the entire metabolomics dataset to display differences between and within groups. For this analysis, MetaboAnalyst 4.0 software was used. Afterwards, to explore the association between the acute metabolic response and muscle hypertrophy, Pearson correlations were calculated between the acute metabolic changes observed for the blood collection times Δ 0–5 and Δ 0–60 min of the first training session (pre-training period) and the increase in the vastus lateralis muscle thickness (Δ muscle thickness).

From the metabolites—the changes in which after the acute session were significantly correlated with changes in muscle thickness—Linear Mixed Models for repeated measures were applied for comparisons between and within groups, assuming participants as a random factor, and group (HL and LL), period (pre- and post-training) and time of blood collection (0 min, +5 min, and +60 min) as fixed factors. Main effects and significant interactions were analyzed using Sidak’s post hoc test. This analysis was performed using PASW statistics software version 18.0 (SPSS, Chicago, IL). Significance was at *p* < 0.05.

## 3. Results

### 3.1. Nutritional Habits

No difference was observed between groups or period (pre, and post-training) for total caloric consumption and amount of macronutrients (carbohydrates, lipids, and proteins) consumed in the days before blood collection. The data are presented in more detail in the Appendix A.

### 3.2. Maximal Dynamic Strength 

A major period (pre vs. post) effect was observed for the 1-RM tests in the 45° leg press and leg-extension exercises (*p* < 0.0001 for both). An increase in the load in the 1RM test was observed after eight weeks of training for both the 45° leg press and the leg extension, when compared with pre-training (*p* < 0.0001 for both) and week 4 (45° leg press: *p* < 0.0001; leg extension: *p* < 0.003), as well as at week 4 compared with the pre-training period (45° leg press: *p* < 0.0001; leg extension: *p* < 0.001). However, there was no between-group difference in the 45° leg press and leg-extension exercises (*p* > 0.05) (Figure 3).

### 3.3. Muscle Activation

The HL group presented greater electromyographic amplitude compared with the LL group, both in the first (*p* < 0.0001) and in the last (*p* = 0.01) training session (Figure 4). Additionally, for the LL group, greater electromyographic amplitude was observed in the last training session compared with the first session (*p* = 0.03); while in the HL group, a trend toward greater electromyographic amplitude was observed in the last training session compared with the first session (*p* = 0.058) (Figure 4).

### 3.4. Muscle Hypertrophy

The main period (pre vs. post) effects were observed and showed an increase in muscle thickness, in both the HL and LL groups, after eight weeks, when compared with the pre-training period for the vastus lateralis thickness (*p* < 0.0001, Figure 5A), sum of the thickness of the rectus femoris muscles and vastus intermedius (*p* < 0.0001, Figure 5B), and sum of the thickness of the vastus lateralis, rectus femoris and vastus intermedius muscles (*p* < 0.0001, Figure 5C). There was no difference between the HL and LL groups for the vastus lateralis thickness (*p* = 0.8), sum of the thickness of the rectus femoris and vastus intermedius (*p* = 0.9), and sum of the thickness of the vastus muscles lateral, rectus femoris, and vastus intermedius (*p* = 0.9). Considering the sum of the thickness of the three muscles analyzed, the HL group showed a percentage increase of 12.5% and the LL group of 12.1%.

### 3.5. Metabolic Response

After the experimental sessions of the HL and LL groups and the performance of ^1^H-NMR spectroscopy, 50 metabolites were identified and quantified in the blood serum in both groups; a table with further details can be found in the Appendix A and the heat map in Figure 6 provides an overview of the levels of each detected metabolite between and within groups.

From the Principal Component Analysis (PCA), it was not possible to show a clear segregation between the HL and LL groups. This finding suggests that there was no global disturbance in the metabolism between and within groups (pre/post-training periods of each group) that could be attributed to a specific group of metabolites (Figure 7).

From the correlational analyses, a significant correlation was observed between the observed changes in the thickness of the vastus lateralis muscle and the metabolic response in the first training session of 0–5 min for the HL group (carnitine: r = 0.678, *p* = 0.045; creatine: r = 0.716, *p* = 0.030), the LL group (acetoacetate: r = 0.715, *p* = 0.046; creatine phosphate: r = −0.620, *p* = 0.018), and the combined groups (carnitine: r = 0.531, *p* = 0.028; creatine: r = 0.532, *p* = 0.028; creatine phosphate: r = −0.517, *p* = 0.033); and 0–60 min for the HL group (3-Hydroxyisovalerate: r = −0.727, *p* = 0.041; phenylalanine: r = −0.750, *p* = 0.032; asparagine: r = −0.721, *p* = 0.044; creatine phosphate: r = −0.772, *p* = 0.025; methionine: r = −0.725, *p* = 0.042), the LL group (oxypurinol: r = 0.821, *p* = 0.045), and the combined groups (carnitine: r = 0.608, *p* = 0.021; creatine phosphate: r = −0.620, *p* = 0.018).

Univariate analyses were then performed for the comparison between and within groups of the metabolites that were correlated with the increase in the thickness of the vastus lateralis muscle.

As a result, an increase was observed in the post-training period compared with the pre-training period in the serum levels of 3-hydroxyisovalerate for both groups (HL: *p* < 0.0001; LL: *p* < 0.0001) with no difference between the groups. Phenylalanine levels increased only in the HL group (*p* = 0.00003), but there was no between-group difference. On the other hand, decreased serum asparagine levels were observed in the post-training period compared with the pre-training period only in the LL group (*p* = 0.003), but it was no different from the HL group.

No significant changes were observed in the levels of the metabolites acetoacetate, carnitine, creatine, creatine phosphate, methionine, and oxypurinol when comparing groups or in the post- compared with pre-training period.

## 4. Discussion

The study aimed to analyze the hypertrophy, muscle activation, and acute global metabolic response of the HL and LL groups. We also sought to correlate the muscle activation and metabolic response data with changes in muscle hypertrophy after eight weeks of training. Our main findings can be summarized as follows: (a) there was greater muscle activation of the vastus lateralis in the HL group compared with the LL group in the first and last training sessions; (b) there were no between-group differences in the metabolic responses; (c) there was correlation between growth of the vastus lateralis in the HL group and levels of the metabolites carnitine and creatine (positive correlations, for delta 0–5 min), 3-hydroxyisovalerate, phenylalanine, asparagine, creatine phosphate and methionine (negative correlations, for delta 0–60 min); (d) there was correlation between growth of the vastus lateralis in the LL group and metabolite levels for delta 0–5 min of acetoacetate metabolites (positive correlation), creatine phosphate (negative correlation) and for delta 0–60 min of oxypurinol (positive correlation); (e) there was correlation between vastus lateralis muscle growth in both groups and metabolite levels for delta 0–5 min of carnitine (positive correlation), creatine (positive correlation), creatine phosphate (negative correlation), and for delta 0–60 min of carnitine (positive correlation) and creatine phosphate (negative correlation).

Jenkins, Miramonti, Hill, Smith, Cochrane-Snyman, Housh and Cramer [8] analyzed muscle activation in untrained individuals during high- and low-load RT. In contrast to our results, they observed no differences in muscle activation after 3 or 6 weeks of training between the groups that trained with 80 and 30% of 1RM in the leg-extension exercise. However, the group that trained with 80% of 1RM showed a significant increase from baseline to weeks 3 and 6. We measured EMG only in the vastus lateralis muscle, whereas in Jenkins, Miramonti, Hill, Smith, Cochrane-Snyman, Housh and Cramer [8] the EMG signal was averaged across the vastus lateralis, rectus femoris and vastus medialis muscles. We also used a multi-joint exercise (leg press) while they used a single-joint exercise (leg extension). The recruitment pattern and mechanical differences between the muscles and exercises may have contributed to the discrepancy in results between our study and Jenkins, Miramonti, Hill, Smith, Cochrane-Snyman, Housh and Cramer [8].

Although EMG has traditionally been used to make inferences about muscle activation, some limiting factors of this method have already been discussed [30]. For instance, it is possible that some motor units reduce their firing rates or become momentarily silent to recover from fatigue [30,31,32]. These fatigued motor units would be replaced by more fatigue-resistant motor units, to maintain sufficient force and to continue the exercise [33,34,35]. Therefore, the greater amplitude of EMG, which could be understood as greater muscle activation (i.e., more motor units active), may mean that this analysis is not sensitive enough, as it represents a “snapshot” of the exercise and does not provide information regarding the total number of motor units recruited during the complete set of exercises. Therefore, even with a greater EMG amplitude in the HL group, as observed in our experiment, it is still possible that the same total amount of muscle fibers were recruited during HL and LL exercise performed until concentric failure.

It is often thought that metabolite accumulation is a stimulus that can influence muscle hypertrophy [17,18,19], and that low-load RT can cause greater or longer-lasting metabolic stress compared with high-load RT [16,36]. Thus, even if the mechanical stimulus is lower during low-load RT, similar hypertrophy to that of high-load RT could be achieved. In fact, some studies showed lower intramuscular metabolic stress (creatine phosphate, pH, or blood lactate concentration) in high-load compared with low-load RT with [22,23,37] or without blood flow restriction [38]. However, in a study where ^1^H-NMR analysis of blood plasma was performed to provide a more comprehensive observation of acute responses to high-load and low-load RT with blood flow restriction, there was no difference in the metabolic stress [24].

Although our results showed no difference in the global metabolic response between the HL and LL protocols, it was possible to observe a relationship between metabolic response and muscle hypertrophy in both groups. Some metabolites correlated with the increase in vastus lateralis muscle thickness (i.e., muscle hypertrophy) in the HL and LL groups. Some of these observed correlations may be associated with the characteristics of the activated muscle fibers (type I or type II) and the metabolic demand of the training protocols used in our study. During HL, type II muscle fibers may be predominantly recruited. These muscle fibers have low oxidative but high glycolytic activity [39,40] and may be more responsive to hypertrophy compared with type 1 muscle fibers [39,41]. On the other hand, LL may preferentially activate type I muscle fibers, which have low glycolytic but high oxidative capacity, and are highly fatigue-resistant [39,40].

In our study, we performed metabolomic analysis in two periods (i.e., pre- and post-training) to examine whether the acute global metabolic response changed after a period of HL or LL. As a result, when looking at the PCA plot and heat map, there was no segregation between groups or within groups pre- and post-training. The clusters formed on the heat map present a grouping when observing the period (pre- and post-training) and time of collection (0, 5, and 60 min) between the HL and LL groups, suggesting that there was no difference in the global metabolic response between the HL and LL groups, or between pre- and post-training for any group. Univariate analysis showed differences only in changes in 3-hydroxyisovalerate (both groups), asparagine (LL group) and phenylalanine (HL group) from pre- compared with post-training, with no between-group difference.

The study by Gehlert, Weinisch, Romisch-Margl, Jaspers, Artati, Adamski, Dyar, Aussieker, Jacko, Bloch, Wackerhage and Kastenmuller [21] had already analyzed the metabolic response after acute resistance exercise and after an additional period of resistance training. Fourteen young male subjects with moderate resistance training experience participated in this experiment, six of whom provided sufficient muscle samples for metabolomics analysis. The first training session altered 33 metabolites, including increases in 3-methylhistidine, N-lactoylvaline, xanthosine, NAD (N1-methyl-2-pyridone-5-carboxamide), and chenodeoxycholate bile acid. The authors also compared metabolite levels after the last training session (i.e., after the training period) with those of the first training session. The comparison showed that only five of the metabolites that had changed after the acute exercise in the first training session were also changed after the last training session. However, it is important to emphasize that the study by Gehlert, Weinisch, Romisch-Margl, Jaspers, Artati, Adamski, Dyar, Aussieker, Jacko, Bloch, Wackerhage and Kastenmuller [21] did not analyze the metabolic response during low-load training, but only the response in high-load training (8 to 12 maximum repetitions). Thus, it is not possible to compare low- and high-load exercise.

We chose to discuss six metabolites, as they may be related to muscle fiber properties and metabolic demand. These metabolites were: asparagine, 3-hydroxyisovalerate, acetoacetate, carnitine, creatine, and creatine-phosphate. Asparagine may have a precursor function as an energy substrate of the Krebs Cycle; it is an amino acid that can be converted into aspartate, then into oxaloacetate for use in the Krebs Cycle [42]. 3-hydroxyisovalerate is a by-product of the leucine degradation pathway, formed from 3-methylcrotonyl-CoA. The leucine degradation pathway leads to the production of acetyl-CoA, another substrate for the Krebs Cycle [43]. Interestingly, these two Krebs cycle energy substrate precursor metabolites showed a negative correlation with the increase in vastus lateralis muscle thickness for the time 0–60 min in the HL group. We hypothesize that the negative correlation was due to stimulation of a greater number of type II fibers, resulting in lower expression of metabolites related to the oxidative system, such as 3-hydroxyisovalerate and asparagine, and concomitantly greater hypertrophy due to the predisposition of type II muscle fibers to hypertrophy [39,41]. Carnitine is also related to oxidative energy production, for which a positive correlation was observed between the changes in time 0–5 min with the increase in vastus lateralis muscle thickness in the HL group. Carnitine has the function of transporting long-chain fatty acids to the mitochondria, which are later oxidized for energy production [44]. Thus, higher availability of carnitine in the circulation could indicate a lower requirement for fatty acid oxidation due to a lower activation of type I fibers, which would explain the positive correlation of this metabolite with the increase in vastus lateralis muscle thickness in the HL group due to a predominant activation of type II fibers.

The concentration of acetoacetate may be another indicator of lower activity of the oxidative system and type I fibers. A positive correlation was observed between the increase in muscle thickness of the vastus lateralis in the LL group and acetoacetate in the time 0–5 min. This metabolite is a ketone body formed from some amino acids and free fatty acids and serves as an alternative source of energy substrate for peripheral tissues, such as skeletal muscle, under conditions of reduced carbohydrate availability [45,46]. In muscle, the β-hydroxybutyrate metabolite produced in the liver is re-oxidized to acetoacetate by the action of the enzyme 3-hydroxybutyrate dehydrogenase. Sequential reactions using acetoacetate lead to the formation of two acetyl-CoA molecules, which are incorporated into the Krebs Cycle for terminal oxidation and adenosine triphosphate (ATP) production [45,46]. The activity of the enzyme 3-hydroxybutyrate dehydrogenase is highest in type I fibers and lowest in type II fibers [45,47]. Therefore, higher acetoacetate levels may indicate lower oxidative system activity and of the type I fibers.

In addition to the metabolites related to the oxidative system of energy production, some metabolites of the alactic anaerobic system showed a correlation with HL and LL. For creatine, a positive correlation was observed with the increase in the vastus lateralis muscle thickness in the HL group at 0–5 min. For the creatine-phosphate metabolite, a negative correlation was observed with the increase in the muscle thickness of the vastus lateralis in the LL group for the time 0–5 min. Creatine in its phosphorylated form (i.e., creatine-phosphate) provides an energy reserve for rapid ATP regeneration, especially during exercise, with high recruitment of type II fibers [39,48]. In addition to the correlation with HL, there was a period main effect, with an increase in plasma creatine levels at post- compared with pre-training. This increase may be related to muscle growth and, consequently, to the reserve of this intramuscular metabolite, which is later released into the bloodstream after performing RT. The concomitant muscle growth and increased intramuscular creatine levels have been demonstrated in previous studies, especially in experiments analyzing creatine supplementation [49,50,51]. 

Phenylalanine and methionine showed a negative correlation with the muscle thickness of the vastus lateralis of the HL group for the time 0–60 min. This correlation may be partly explained by the characteristic incorporation of these metabolites into synthesized proteins. Both phenylalanine and methionine are known to be used as tracers to measure the rate of protein synthesis, owing to their incorporation during protein synthesis [52,53,54,55]. In contrast to the above metabolites, which appear to correlate with muscle hypertrophy based on muscle fiber properties and metabolic demand, phenylalanine and methionine may act as regulators of protein synthesis. Increased phenylalanine release from skeletal muscle has been shown to be associated with decreased mTOR activation and a concomitant decrease in cell growth signaling [55]. Methionine is a methyl group donor involved in DNA and protein methylation through the formation of S-adenosylmethionine, which is a known indirect activator of the mTOR pathway [56,57]. Interestingly, the correlation of these metabolites was observed only for HL; we cannot explain this finding.

Finally, the metabolite oxypurinol showed a positive correlation with the muscle thickness of the vastus lateralis in the LL group at time 0–60 min. This metabolite is an inhibitor of xanthine oxidase, an enzyme that generates reactive oxygen species (ROS) that converts hypoxanthine to xanthine [58]. During the execution of RT, a greater amount of ATP is consumed than can be resynthesized in the exercising muscles, which may lead to the depletion of adenosine diphosphate to maintain force production, increasing the plasmatic concentrations of hypoxanthine and, consequently, xanthine and ROS [58]. Chronically, the formation of ROS may have negative effects on muscle tissue or even induce sarcopenia [59]. Thus, a possible explanation for the positive correlation of oxypurinol with the muscle thickness of the vastus lateralis of the LL group would be that individuals with higher oxypurinol production had lower formation of ROS throughout the training period and consequently greater muscle growth compared with individuals with lower oxypurinol levels.

## 5. Conclusions

Performing high- and low-load RT to concentric failure induced a similar global metabolic response and increased the strength and muscle thickness of the quadriceps muscles analyzed, although the HL showed greater muscle activation. However, correlations were observed between some metabolites from the first training session and the increase in muscle thickness in the HL and LL groups. Some correlations may be related to the characteristics of the activated muscle fibers and the metabolic demands of the training protocols used (3-hydroxyisovalerate, asparagine, acetoacetate, carnitine, creatine, and creatine-phosphate), while the others may be related to the process of protein synthesis (phenylalanine and methionine). The observation of these correlations with the increase in muscle thickness may be of great importance, as these metabolites may be used as biomarkers for muscle hypertrophy in the future. However, these results should still be interpreted with caution, and further studies are necessary to investigate the efficacy of using metabolites correlated with muscle growth as biomarkers.

## 6. Limitations

Our study was not without limitations. We collected blood samples in pre- and post-training, at three time points (0 min, 5 min, and 60 min). It is possible that with more data points, we could have described a better time course of metabolic response. However, it is difficult to collect more than three blood samples from participants on the same day, after overnight fasting and 1 h after the exercise. We performed metabolomic analysis on blood plasma only. It would be interesting to analyze the level of intramuscular metabolites. However, we could not obtain muscle samples by muscle biopsies. Nevertheless, it is important to highlight that the use of blood serum, although a limitation of the study, may serve as a basis for the identification of new biomarkers to evaluate skeletal muscle adaptation by developing a simple and less invasive procedure in the future compared with muscle samples.

## Figures and Tables

**Figure 1 metabolites-13-00335-f001:**
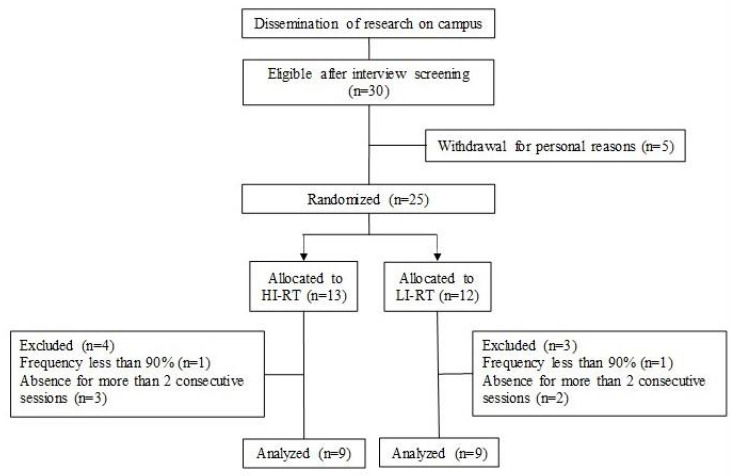
Flow diagram of participants in the study.

**Figure 2 metabolites-13-00335-f002:**
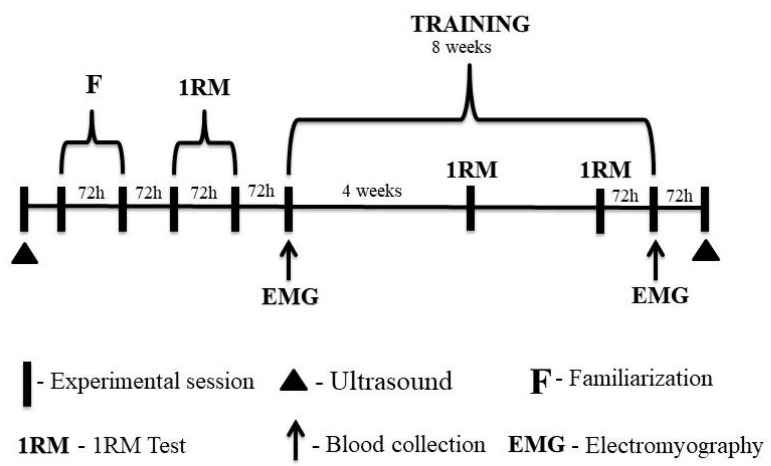
Experimental design of the study.

**Figure 3 metabolites-13-00335-f003:**
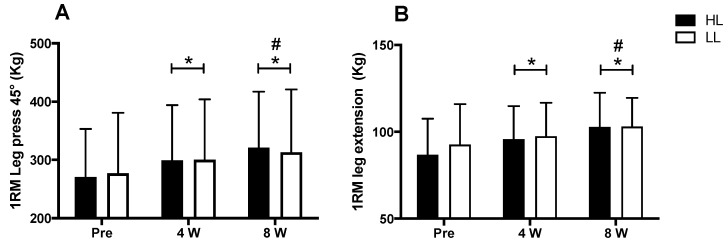
1RM test in leg press 45° (**A**) and leg-extension exercises (**B**). HL: high-load resistance training, LL: low-load resistance training, Pre: test performed before eight weeks of training, 4 W: test performed in the fourth week of training, 8 W: test performed in the eighth week of training before the last training session. * *p* < 0.05 when compared with Pre. # *p* < 0.05 when compared with 4 W.

**Figure 4 metabolites-13-00335-f004:**
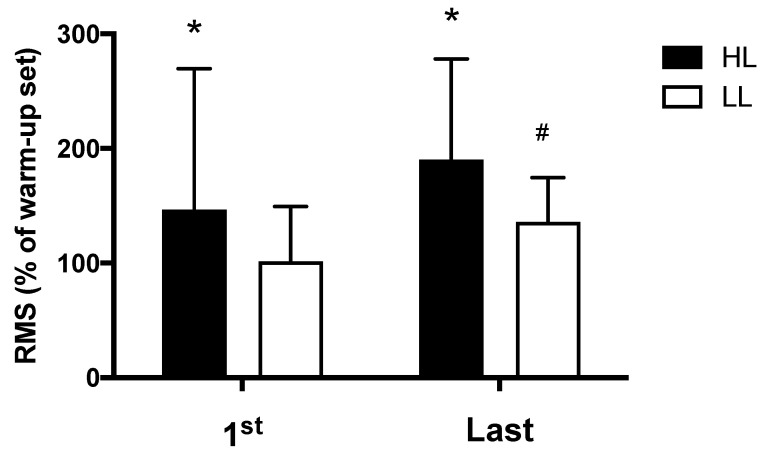
Amplitude of the electromyographic signal, RMS: normalized root mean square (percentage of the warm-up set used for normalization), HL: high-load resistance training, LL: low-load resistance training, 1st: first training session, Last: last training session. * *p* < 0.05 compared with the LL at the same period. # *p* < 0.05 compared with 1st for LL.

**Figure 5 metabolites-13-00335-f005:**
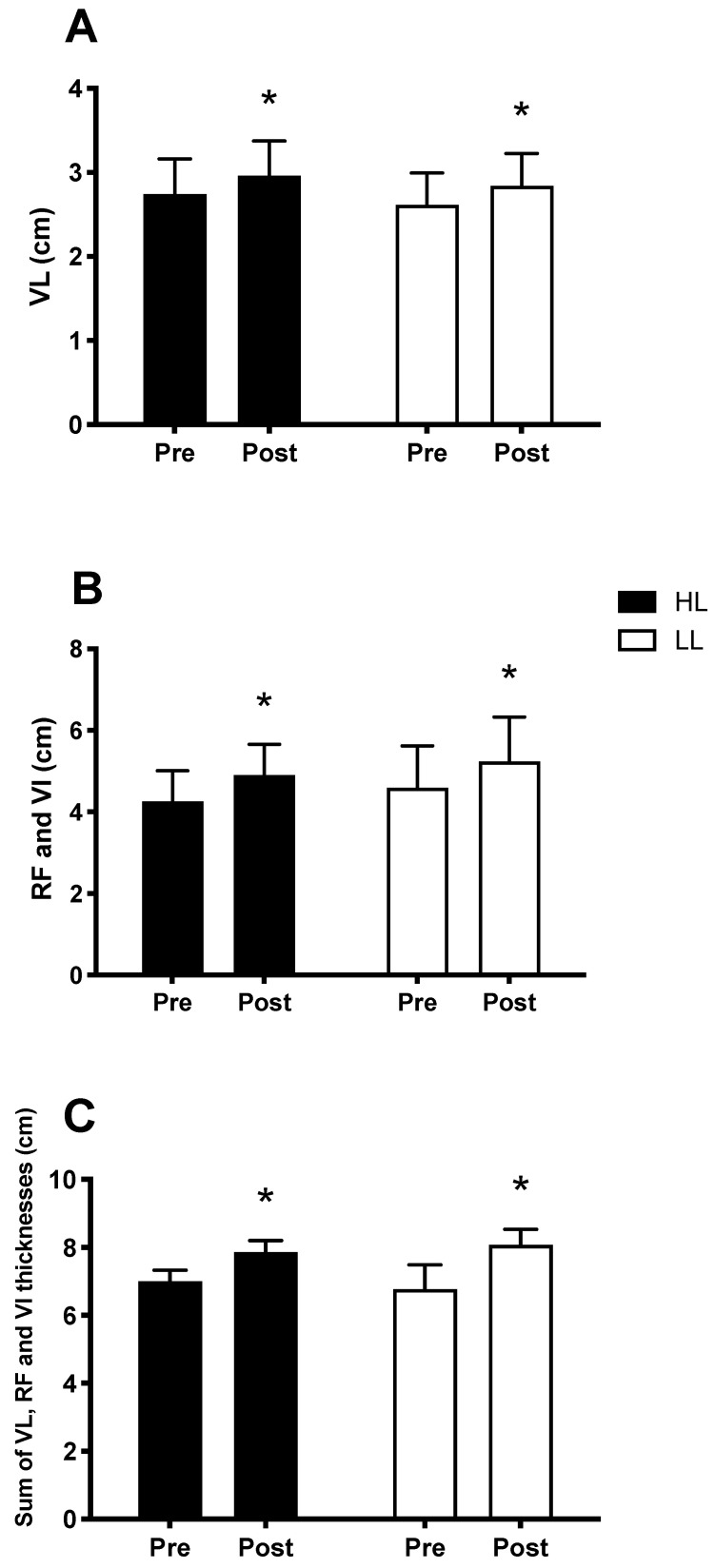
Thickness of the vastus lateralis muscle (VL) (**A**); sum of the thickness of the rectus femoris (RF) and vastus intermedius muscles (VI) (**B**); sum of the thickness of the vastus lateralis, rectus femoris and vastus intermedius muscles (**C**); Pre: before the training period, Post: after eight weeks of training. HL: High-load resistance training, LL: low-load resistance training. * *p* < 0.05 compared with the Pre intra-group period.

**Figure 6 metabolites-13-00335-f006:**
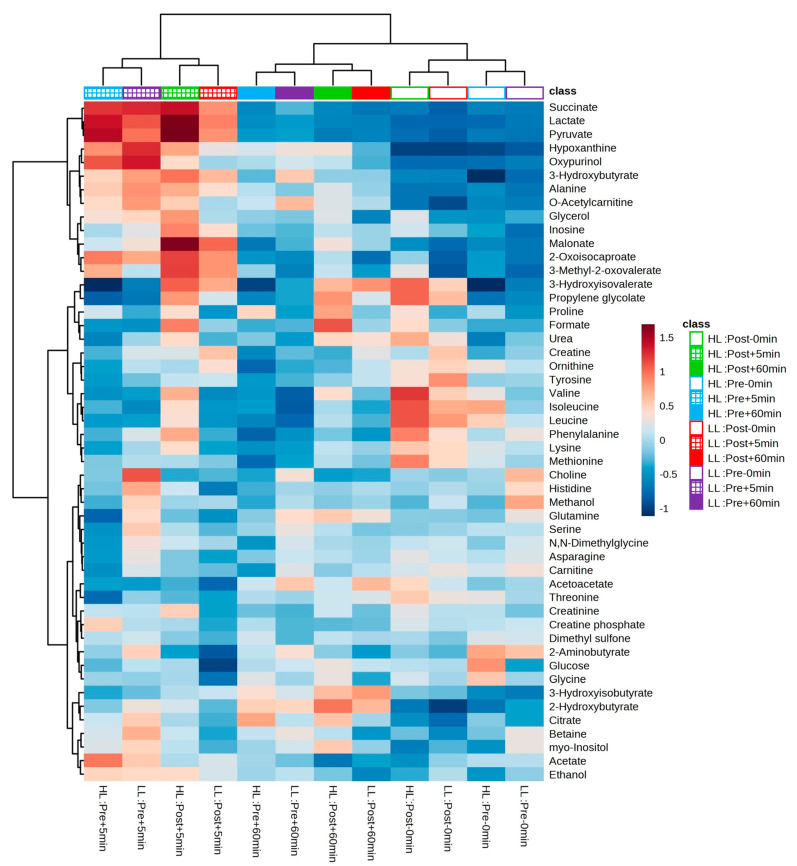
Heatmap and hierarchical clustering analysis (based on the Euclidian distance measure and the Ward clustering algorithm) between and within groups. Cells represent the mean of standardized concentration levels (auto-scaled data). Blue represents lower concentration level, and red represents higher concentration level. HL = high-load resistance training; LL = low-load resistance training; Pre = pre-training period; Post = post-training period; −0 min = blood collection performed before the exercise (baseline); +5 min = blood collection performed 5 min after exercise; +60 min = blood collection performed 60 min after exercise.

**Figure 7 metabolites-13-00335-f007:**
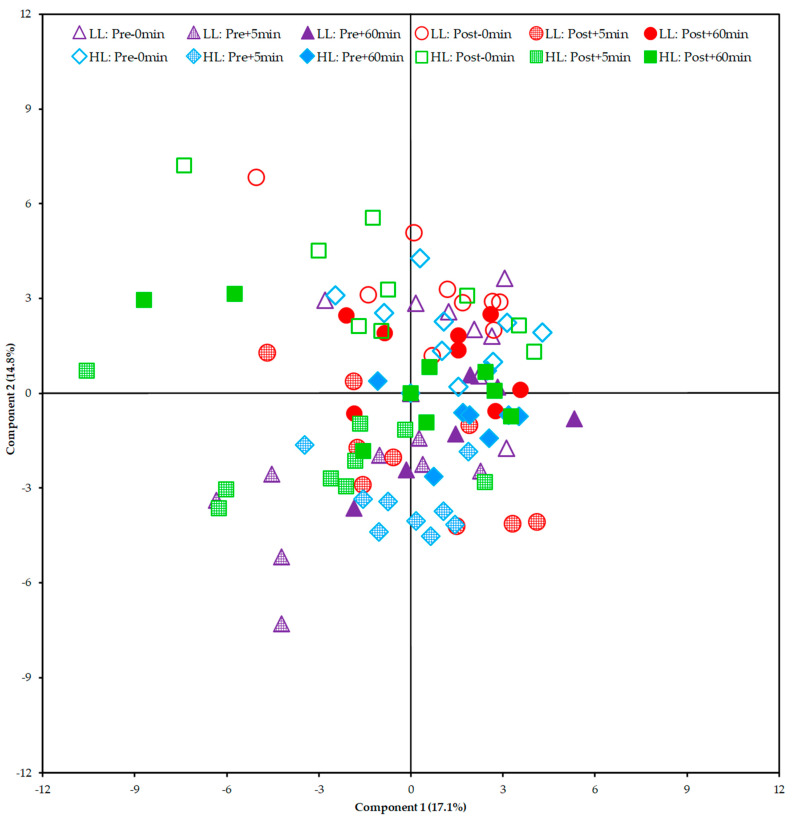
Score plot of the Principal Component Analysis (PCA) including all samples. HL = high-load resistance training; LL = low-load resistance training; Pre = pre-training period; Post = post-training period; −0 min = blood collection performed before the exercise (baseline); +5 min = blood collection performed 5 min after exercise; +60 min = blood collection performed 60 min after exercise.

## Data Availability

Not applicable.

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
