# Peer review of "Serum Metabolites Associated with Muscle Hypertrophy after 8 Weeks of High- and Low-Load Resistance Training"

_metabolites, 2023, doi:10.3390/metabo13030335_

Round 1

Reviewer 1 Report

The authors studied the acute metabolite changes after high- and low-load resistance training and its correlation with muscle hypertrophy in untrained young men. Although the experimental design is clear, there are many important issues that must be taken into consideration for improving the manuscript. For instance, the authors measured metabolic alteration in the blood and not in within muscle making more difficult to interpret the changes. In addition, a major flaw of the study is the lack of novelty in the field. There are many mechanistic studies that aimed to correlate alterations in the metabolites within the working muscle with changes in activation and important signaling pathways that culminate in hypertrophy. Finally, the metabolomic data is not well explored and showed in a better manner.

Reviewer 2 Report

The present study investigates the muscle activation and the overall metabolic response of acute sessions, using metabolomics, of high- (HL) and low-load (LL) RT before and after a chronic period of training. Furthermore in the study muscle thickness by ultrasound and sEMG amplitude by electromyography is measured. The hypertrophy, muscle activation, the acute global metabolic response of the HL and LL is analyzed and compared under resepected of acute and chronic adaptation. It is demonstrated that HL and LL RT until concentric failure induced a similar global metabolic response, increased strength and muscle thickness of the quadriceps muscles, despite the HL presenting greater muscle activation. Correlations were observed between some metabolites from the initial training session and the increase in muscle thickness in HL and LL.  Correlations of metabolites with activation of muscle fibers and dependent metabolic demands of the used training protocols as well as to the process of protein synthesis are found. Although the study is principal interesting a major limitation  is given by the lack of measurement of classic metabolic parameter as oxygen demand and lactate production.

Specific comments:

1.      It would be interesting to get the training energy expenditure in HL and LL groups.

2.      It is surprising that no lactate measurement is done during the RT. Moreover interpretation of the metabolism would also be improved if spiroergometry would be performed.

3.      By which criteria the selection of the six metabolites are done?

4.      It would be interesting to normalize the given results by muscle mass of the used muscle groups during training.

Round 2

Reviewer 1 Report

The authors made an effort to include new graphic and show the heat map. However, the major flaw of the study is still the lack of novelty in the field.

Reviewer 2 Report

no further comments

Author Response

We would like to thank the reviewer for his time and effort in reviewing our manuscript.